# Natural History and Exploitation of the Harbor Porpoise (*Phocoena phocoena* Linnaeus, 1758) during the Neolithic (ca. 4000–2000 cal. BC) in the Eastern Baltic Region

**DOI:** 10.3390/ani13050909

**Published:** 2023-03-02

**Authors:** Lembi Lõugas, Valdis Bērziņš

**Affiliations:** 1Archaeological Research Collection, Tallinn University, EE-10130 Tallinn, Estonia; 2Institute of Latvian History, University of Latvia, LV-1050 Riga, Latvia

**Keywords:** harbor porpoise, eastern Baltic Sea, Neolithic, ceramics, Littorina Sea

## Abstract

**Simple Summary:**

The harbor porpoise, a small marine mammal related to dolphins and whales, is nowadays very rarely seen in the eastern part of the Baltic Sea, and was not common here in recent centuries. However, the bones of this animal have been found on many Stone Age archaeological sites dating from the period 4000–2000 BC along the coasts of Estonia, Latvia and Lithuania, which shows that it was frequently being hunted by the coastal communities of that time. In this period, the Baltic Sea was saltier than it is today, making conditions more favorable for marine species such as the porpoise. In addition to meat and blubber, the porpoise also provided a tool for decorating pottery vessels made by the Stone Age inhabitants. This instrument was made from the animal’s jaw bone, the teeth being impressed in the clay to produce a distinctive row of marks. Such decoration on pottery may have symbolized the coastal way of life and joint hunting activities. Pottery with porpoise tooth ornamentation has been found on many coastal Stone Age sites in the region and also in smaller amounts at inland sites, providing evidence of contact between coastal and inland communities.

**Abstract:**

Compared with the history of seals in the Baltic Sea, the porpoise has received much less research attention. The harbor porpoise (*Phocoena phocoena*) has been quite rare in the eastern Baltic in recent centuries, but according to archaeological finds, its population was quite numerous here ca. 6000–4000 years ago (ca. 4000–2000 cal. BC). This paper deals with all known archaeological assemblages of porpoise so far discovered in the eastern Baltic (Estonia, Latvia and Lithuania), discusses the hunting strategies and studies the exploitation of this small cetacean by the Neolithic hunter-gatherers. Fauna historical aspects include new archaeological data in addition to those published previously. We consider whether these new data change the temporal and spatial pattern of porpoise hunting and examine how, in addition to the expected use of porpoise meat and blubber, the porpoise’s toothed mandibles were used for patterning ceramics.

## 1. Introduction

The harbor porpoise (*Phocoena phocoena* Linnaeus, 1758) is one of the world’s smallest cetaceans. As the name indicates, it stays close to coastal areas and is, thus, particularly visible to people interested in whale watching. The porpoise can swim up rivers and be found hundreds of kilometers from the sea. However, marine coastal waters constitute its most common environment, and the porpoise is a widespread species in cooler coastal waters of the Northern Hemisphere. In the eastern North Atlantic, the harbor porpoise is distributed along the coast of the Barents Sea, the west coast and fjords of Norway, the coasts of Iceland and the British Isles, in Danish waters and along the West European coast as far as north-western Africa [1,2]. Our main interest is in its distribution in the Baltic Sea, and more specifically, its distribution in the eastern Baltic Sea, with an almost freshwater environment. Not that porpoises avoid freshwater as such, but the environment seems to be unsuitable for permanent habitation by this species. The reason is most probably the shallowness of the littoral zone, which this pelagic species does not like, and severe ice conditions in winter [3].

The Baltic population of porpoise is not large nowadays. The population inhabiting the Baltic Proper has been classified as ‘critically endangered’, on the basis that the current population is likely to be fewer than 250 individuals [4,5]. Some decades ago it was estimated that a maximum of 600 individuals occurred in the Baltic, mainly in the southern part of the sea [6]. However, various studies and examinations of sighting and stranding data support the general view that numbers of harbor porpoises in the Baltic Sea have declined and that the distribution range has narrowed [3]. Although studies of mitochondrial DNA and microsatellites show that the Baltic population of porpoises forms a separate geographical unit [7], the (seasonal) migrations between the Baltic and Kattegat/North Sea still exist. As there is no difference in the isotopic (^13^C) signature of the porpoises in the Baltic and the Kattegat/Skagerrak, migration between these areas is evident [8]. Despite the low number of porpoises today, this species seems to have been a much more abundant inhabitant in the Baltic in the past.

In the coastal waters of Estonia, Latvia and Lithuania, cetaceans have been rare visitors rather than permanent inhabitants. This applies to porpoise as well as other cetacean species (at least, the four species registered). Only recently, in September 2008, white-beaked dolphins (*Lagenorhynchus albirostris* Grey, 1846) were seen as far east as Tallinn Bay in northern Estonia. On 1 June 2020 a couple of dolphins (most probably bottlenose dolphins) were encountered in Kopli Bay (photographed by the captain of the boat Indrek Sülla), and a humpback whale (*Megaptera novaeangliae* Borowski, 1781) was seen off the coasts of Poland and Finland. Probably the same individual was finally found dead close to Skulte harbor on the Latvian coast of the Gulf of Riga in July 2006 (information from local newspapers). Older records provide more evidence of the presence of whales in the eastern Baltic. Besides humpback whales, white whales (*Delfinapterus leucas* Pallas, 1776) are also mentioned [9].

The porpoise, which has permanently inhabited the Baltic for several millennia, can likewise be considered a very rare visitor in the eastern Baltic Sea today: only one or two proven observations per decade are known. It was not like this in the past. In the late 19th and early 20th century, the porpoise was encountered more regularly here [10] (p. 487), [11] (pp. 56–57), [3] (p. 168). In the 1920s and 30s, the porpoise often made it into the newspapers: Ivar Jüssi (in *Õhtuleht*, 29 September 2012) concluded that there are ca. 40 mentions from this period in Estonia, the best known being a picture of “fishermen from Virtsu with a strange fish” (Figure 1).

In the Middle Ages and Early Modern Era, the porpoise (*merswyn*) is mentioned in accounting books among other animal products sold in Tallinn’s markets [12] (p. 9). The archaeological record of that period includes only one vertebra of porpoise, found in a mixed layer at 6A Tatari Street, Tallinn (identified in 2022). It seems that porpoise was more like an exotic import in towns of the eastern Baltic at that time, rather than being hunted by local people.

In comparison with the history of seals in the Baltic [13] (and references therein), especially the history of the harp seal (*Phoca groenlandica* Erxleben, 1777), the porpoise has received much less research attention. It has been considered as part of the hunted fauna in zooarchaeological studies, but has not been the subject of any special research in taphonomy. Even though the species has been quite rare in the last centuries, according to archaeological finds, its population was quite numerous ca. 6000–3500 years ago. Thus far, only one paper has summarized the post-glacial spatiotemporal distribution of porpoise in the Baltic Proper [14], but with insufficient data on porpoise finds from the eastern Baltic. An investigation by Aaris-Sørensen et al. [15] focuses on finds of all cetaceans, but is geographically limited to the southern Scandinavian coast. Data on finds from the eastern Baltic are scattered in reports and papers having some other focus.
Figure 1Fishermen with a strange fish in Virtsu, West Estonia, in 1939. Photo from the Virtsu Hobby Museum—VHM-F0441 [16].
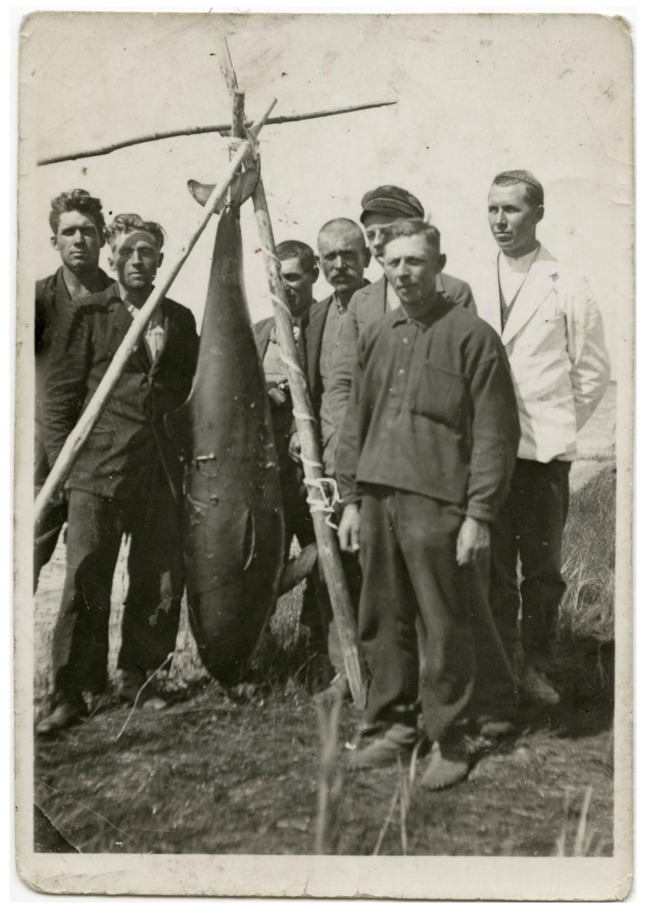



This paper focuses on the archaeological finds of porpoise discovered in the eastern Baltic—Estonia, Latvia and Lithuania—and traces the faunal history of this species as well as its exploitation by humans, covering the period from the saline stage of the Baltic Sea (Littorina Sea) up to the brackish water (Limnea Sea, modern) stage. In cultural terms, the period under consideration corresponds to the New Stone Age or Neolithic. The exploitation of porpoise involved the use not only of hunting products for food but also of bones for practical purposes. Thus, the porpoise mandible with teeth still in place was widely utilized as a tool for decorating pottery.

## 2. Zooarchaeological Material

Our study material for clarifying the history of the porpoise comes from archaeological contexts, as no finds of porpoise from geological contexts are known in Estonia, Latvia or Lithuania. Most of the material has been excavated and collected manually from Neolithic dwelling sites situated on islands, in addition to finds from coastal areas of the mainland (Figure 2), and is dated roughly to the time span 4000–2000 cal. BC.

Porpoise has been identified in the archaeological material of Kunda Lammasmägi by Johannes Lepiksaar [17,18], at Riigiküla I and III, Naakamäe and Loona by Kalju Paaver [19], and in Tallinn (1 Vabaduse Sq.) and at Kõpu XI, Riigiküla IV and Siliņupe by Lembi Lõugas [20,21,22,23,24]. Data on finds from Šventoji have been published by Linas Daugnora [25]. The Naakamäe and Loona bones have been re-examined by Lõugas, since no anatomical division is given in Paaver [19].

The taxonomic and anatomical determinations were performed using comparative skeletal collections and skeletal atlases. In addition, quantitative methods such as the number of identified specimens (NISP) and the minimum number of individuals (MNI) have been used in most cases (Table 1). Since mainly the vertebrae and/or vertebral bodies have been found, the MNI is not always counted, as the vertebrae assemblages provide no certainty as to the number of individuals present.

From the Lammasmägi site at Kunda, which is considered mainly a Mesolithic site, at least one find of porpoise is recorded [17,18] (thereafter Lõugas [26,27]). However, it is not clear which bone this was, and the bone is not accessible or else has been lost from the collection. A brief report by Kalju Paaver (held in the archive of the Archaeological Research Collection, Tallinn University), which includes the results of animal bone identifications based inter alia on the material found at Kunda Lammasmägi in the 1930s, makes no mention of porpoise. Perhaps the find had already been lost by that time. An inventory of bone collections in 2022 did not clarify the existence of the porpoise find either. Thus, some doubt remains as to whether there was any porpoise at the site at all. The Mesolithic context of the site does not support the existence of a porpoise find there, but as has been ascertained earlier, Neolithic finds were also found at the Kunda Lammasmägi site, as confirmed by ^14^C (radiocarbon) dates, and both Narva and Comb ceramics have been recorded [28,29,30]. However, because of the uncertainty, this porpoise record will not be discussed further.

The most exceptional material in terms of porpoise bones come from the Neolithic site of Naakamäe on Saaremaa Island in western Estonia. Here, a huge quantity of marine mammal remains were found in the 1960s. Although the material is dominated by harp seal, porpoise was also intensively hunted [13,19,27]. At Naakamäe, all body parts (skull fragments, vertebrae and limb bones) are represented in the zooarchaeological material, but only four fragments of mandibles are present.

The Loona site is situated quite close to Naakamäe, but is somewhat younger according to the archaeological context and ^14^C dates [26]. Here, a huge amount of cod and seal bone was found. However, porpoise is represented by two vertebrae only [19,20].

North of Saaremaa, another island—Hiiumaa—has also yielded a few porpoise bones. Many seal hunters’ campsites are known on Kõpu, the westernmost peninsula of Hiiumaa [31,32]. The Neolithic site Kõpu XI, situated in the southern part of the ancient Kõpu Island, is similar to Naakamäe and Loona, but in contrast to the Saaremaa sites, the bone material is very fragmented and most of it is burnt. Such preservation makes it difficult to identify bones for taxonomic purposes. However, there are five fragments of porpoise skeleton: burnt temporal bones (i.e., *bulla tympanica*), the hardest part of the skull, which preserves better than other bones of the skeleton. In addition, one unburnt fragment of a vertebral disk may come from a young porpoise, but this assumption is based on the disk’s structure rather than its shape. Namely, the porpoise, like other cetaceans, has a special bone structure, which differs from that of terrestrial mammals. However, as the disk fragment is quite small, some doubt may still remain as to whether it comes from the seal or porpoise (Kõpu XI finds were re-examined by Lõugas in 2022).

Two Neolithic sites on the coast of the mainland—Tallinn Vabaduse Sq. (Estonia) and Siliņupe (Latvia)—are similar in terms of their geographical location and faunal evidence. Both are situated at the southern shore of a bay close to a river estuary and, besides terrestrial animals, have yielded considerable amounts of harp seal and porpoise bone [23,24]. It is worth mentioning that only one ca. 6 cm fragment of a mandible is recorded at the Vabaduse Sq. site, and none at Siliņupe. The mentioned geographical locations correspond to the old hunting method of driving the migrating porpoises into narrow and shallow bays, where hunters can slaughter them [33]. 

The other sites considered here, Riigiküla I, III and IV in north-eastern Estonia and Šventoji 2B and 3B in western Lithuania, are not very rich in porpoise bones [19,21,25]. Only a few vertebrae have been found, although the hunting of harp seal was apparently also practiced by the settlers of Riigiküla and Šventoji. The Riigiküla finds are important, since they prove the occurrence of porpoise as far as the eastern part of the Baltic Sea, in the eastern Gulf of Finland.

There is also one lumbar vertebra of porpoise from Tallinn found in mixed layers of medieval/modern age at 6A Tatari Street. In the absence of ^14^C dates, it does not provide definite confirmation of the occurrence of porpoise in the mentioned period. Thus, 6A Tatari Street is not far from the Vabaduse Sq. site (ca. 200 m), where the Stone Age finds are partly mixed with the medieval and early modern finds. Although Stone Age layers are absent in Tatari Street, a few finds indicating the Stone Age are known in the vicinity [34]. Currently, in the absence of ^14^C dates and according to the archaeological context, we can relate this porpoise find to the medieval and/or modern period.

## 3. Porpoise Tooth Impressions as Decoration on Pottery

In addition to the faunal remains testifying to the porpoise’s importance as a source of meat and blubber for coastal communities in the Neolithic, a recent study has revealed that porpoise hunting also yielded a tool used to create a distinctive kind of pottery decoration. As described by Bērziņš and Dumpe [35], the elongated comb-like impressions consisting of a row of fine lenticular pits (Figure 4), previously documented as “fine teeth” [36] or as a special kind of comb decoration [37], are, in reality, impressions made with a row of porpoise teeth.

Experiments using a harbor porpoise mandible (Figure 3) demonstrated convincingly how such impressions could be achieved using one half of the porpoise mandible. The ramus of the mandible provides a convenient handle, and a stamp with the desired number of teeth could be prepared by simply removing the superfluous teeth from the mandible. Since the dentition of the porpoise differs radically from that of all other members of the region’s fauna, there is no possibility of confusion with tooth impressions of other animals. In cases where the imprints are clear, they may be unequivocally distinguished from impressions made with, for example, a denticulated shell edge or some other specially prepared stamp.

The row of tooth imprints may be straight or somewhat curved. In general, the orientation of the long axis of individual tooth imprints tends to follow the overall direction of the row of imprints, although in many cases, individual teeth are somewhat obliquely oriented. In cases where a row of somewhat obliquely oriented porpoise teeth has been used to make a relatively deep imprint in the clay, it may resemble a cord impression; however, in such cases, the tooth imprints will be deeper than imprints left by windings of cord. In cases where the stamp has made a shallow impression, there will be short linear marks left by the tips of the teeth, whereby the overall impression of the stamp resembles a “dashed line” (Figure 3b).

Porpoise tooth decoration was first recognized on pottery from the Sārnate site, near the open-sea coast of north western Latvia, and has since been documented in the pottery assemblages from a number of eastern Baltic coastal habitation sites dated to the final part of the Stone Age. These include sites on the western shore of the Gulf of Riga (Ģipka A and B, lower layer of Pūrciems C) and at the head of the gulf (Siliņupe and Slocene), all in present-day Latvia, as well as the above-mentioned Estonian sites of Vabaduse Sq. and Naakamäe (for the locations, see Figure 2).

The pottery with porpoise tooth impressions from Latvia’s coastal belt is classed as Early Sārnate Ware. Five ^14^C dates from Siliņupe indicate a time span of 3640–2935 cal. BC for the Early Sārnate Ware phase at this site, which has also yielded abundant faunal remains of porpoise [24]. Three dates from dwellings with Early Sārnate Ware at Sārnate itself give an interval of 3516–2880 cal. BC [38]. One ^14^C date has now also been obtained for the lower layer of Pūrciems dwelling C, which has likewise produced Early Sārnate Ware, indicating the time interval 3636–3382 cal. BC (wood charcoal from the 1936 excavation; 4761 ± 30 BP; FTMC-UM96-6; 95.4% probability). 

The Vabaduse Sq. and Naakamäe sites in Estonia yielded pottery classed as Typical Comb Ware and Late Comb Ware [37,39]. The ^14^C dates from these two sites place them in the time interval 3300–2800 BC [20,23,40]. Although ceramics from Estonian coastal sites have been thoroughly studied, experiments to replicate the impressed decoration have only been carried out using a specially made comb-like tool.

Comprehensive study has been undertaken on the pottery from Sārnate, Siliņupe and Riņņukalns. Porpoise tooth decoration is found on the exterior of 28% of the separate vessels identified from rim sherds recovered from dwellings of the Early Sārnate Ware phase at Sārnate, and on 9% of the vessels from area 7 at Siliņupe (Appendix A). 

The porpoise tooth stamp impressions in the Sārnate assemblage have an overall length of up to 29 mm, with imprints of four to eight individual teeth; at Siliņupe, the overall length reaches 23 mm, with imprints of four to eight teeth. 

The pottery with decoration now identified as porpoise tooth, classed as belonging to Early Sārnate Ware in Latvia [41] and to Late Comb Ware in Estonia [37], is viewed as manifesting a fusion of the Narva tradition of pottery-making, represented in the eastern Baltic already before 5000 cal. BC, and the Comb Ceramic tradition, which had spread to the region from the north-east at ca. 4000 cal. BC [41] (pp. 40–41), [42]. The pottery with porpoise tooth decoration, used primarily for cooking [41] (p. 142), generally has a porous fabric with lamellar voids remaining from shell temper that has been leached away (the shell is still present in cases where the pottery has lain in a calcareous substrate, e.g., at Riņņukalns, Figure 4: 4, 7). The designs on vessel exteriors made with porpoise tooth stamps include horizontal rows of vertically or obliquely oriented impressions, rhombuses, herringbone patterns and zigzags, sometimes in combination with shallow pits (Figure 4). These designs essentially belong to the decorative repertoire of Comb Ceramics, the porpoise tooth stamp being utilized in place of a comb or cord stamp.

Considering that the porpoise could only have been hunted at the coast and along the lower courses of rivers, a very significant discovery is the occurrence of porpoise tooth impressions also on pottery from inland sites; it has been found at Riņņukalns, located at the outlet of the River Salaca from Lake Burtnieks. At this site, 95 km upriver from the coast, porpoise tooth decoration occurs on the exteriors of 6% of the vessels assigned to the phase of midden accumulation (Appendix A), dated to 3400–3200 cal. BC [42]. 

A number of pottery sherds with such decoration are also known from approximately contemporaneous ceramic assemblages in the area around Lake Lubāns [35], which would have been reached from the coast by traveling more than 200 km up the Rivers Daugava and Aiviekste. In this intensively researched area, porpoise tooth impressions have so far been identified on four sherds from Malmutas grīva (Figure 4: 5), two sherds from Piestiņa (Figure 4: 2) and 14 sherds derived from at least three vessels in the assemblage from Suļka. In all cases, porpoise tooth decoration occurs on pottery with characteristic lamellar pores indicative of shell temper. In this inland region, porpoise tooth is rarely encountered in comparison with other stamp forms, although it may be somewhat more common than it appears to be, because secure identification is not possible in the case of weathered or degraded sherds. 

As far as can be ascertained from the highly fragmented material, the designs executed in porpoise tooth stamps on the pottery vessels found at inland habitations resemble those seen on pottery from coastal sites. Various other similarities to the coastal pottery, such as the occasional occurrence of knot, plait and beaver metapodial impressions, indicate that some of the pottery occurring on the inland sites was made by people whose pottery-making techniques corresponded to those of the inhabitants of the coastal belt.

## 4. Discussion

### 4.1. The History of the Porpoise in the Eastern Baltic Sea

According to Sommer et al. [14], the harbor porpoise is first encountered on the southern Baltic settlement sites of the Maglemose and Kongemose Cultures, i.e., ca. 7600–5000 cal. BC. However, as the opening of the Danish Straits and the inflow of saline water into the Baltic Proper took place after 7000 BC [43], porpoise could spread to the Baltic only after this geological event. Unfortunately, most of the early finds of porpoise considered in Sommer et al. [14] were dated according to the archaeological context and not by using direct ^14^C dating (as in our study). The only exception seems to be a porpoise from the Stora Förvar site on the island of Gotland, Sweden, which has been dated to ca. 4080 cal. BC. This seems to be in accordance with the dates of the first harp seals found on Baltic settlement sites [13]. Both species appear on the Neolithic settlement sites of the eastern Baltic approximately at the same time, and on all the sites where bones of porpoise were recorded, harp seal occurs in higher numbers.

In the eastern Baltic, the Riigiküla site provides the oldest archaeological context with porpoise bone finds, relating to the Narva and Typical Comb Ware cultures, i.e., ca. 4000 cal. BC [44]. The association of the porpoise finds with the dated material is not very certain, and more evidence, such as a direct ^14^C date, is needed. The highest number of bones of this cetacean relates to the time span 3400–3000 cal. BC (Late Comb Ware, Early Sārnate Ware). This coincides with the period of highest salinity in the Baltic [43], probably offering a suitable environment for marine fauna. The abundance of porpoises in the eastern Baltic diminished after 3000 BC, when the salinity in this sea also decreased, since the outflow of brackish Baltic Sea water became more intense than the inflow of saline water from the Atlantic, and subsequently the visits by this small cetacean became rare. This assumption is also based on the evidence of faunal material from the Asva and Ridala sites (ca. 800–500 BC) on Saaremaa Island, where intensive seal hunting took place but no porpoise bones have been found so far [19,21,27,45]. Certainly, more ^14^C-datings of porpoises are needed from the Baltic region in order to trace the history of the harbor porpoise in this sea. 

### 4.2. The Role of the Porpoise in Human Lifeways

In the middle and second half of the 4th millennium BC, porpoise tooth decoration came to be commonly applied on pottery in the eastern Baltic region. One half of the porpoise mandible, with its long, straight row of teeth, could be modified with minimal effort so as to furnish a pottery stamp comparable in length and width to the comb stamps that had already been very widely used to decorate pottery in the region for several centuries before this. Hence, impressed decoration with porpoise teeth could readily be accommodated within the established design grammar of pottery ornamentation, the porpoise tooth stamp taking the place of the comb stamp. However, we may also ask—given the porpoise’s economic importance for certain coastal communities in this period—whether the stamping of pottery with an instrument fashioned from part of this animal also had a special meaning related to the animal’s salience in the consciousness of the pottery-makers. 

There is practically no other evidence to suggest that the porpoise held a prominent position in the world-view of the Neolithic people in this region. In contrast to the large number of terrestrial mammal and bird representations in the form of figurines and zoomorphically carved artefact parts [46], so far, just one piece has been interpreted as a representation of a porpoise. This is a 36 mm long clay figurine from the north Estonian site Jägala Jõesuu V, which appears to display characteristic porpoise features, notably the dorsal fin [47]. In connection with this, we may note that the other marine mammals hunted in the region, namely seals, are likewise virtually absent from mobiliary art. On the other hand, seals are quite commonly represented by the use of their teeth for making tooth pendants, as found in graves at the Zvejnieki cemetery [48] (pp. 83–89), [49] (pp. 144–145), whereas the excavated burials in the eastern Baltic have not yielded any body parts of porpoise. 

It may actually be that decoration on pottery served as the main medium for expressing the value that coastal communities attached to the porpoise. Because of the distinctiveness of this animal’s teeth and the imprints obtainable from them, such a message could be conveyed quite unambiguously. It is also possible to delineate the range of mental associations that the porpoise could have had. In the first place, of course, it would have been connected with the coastal and river-mouth environments that it inhabits. Further, the historical record indicates that a very important method of porpoise exploitation was mass hunting by coordinated drives, involving several boat crews with nets, as on the coast of Denmark, for example [50]. If similar collective methods were being employed in antiquity, then the joint endeavors of porpoise hunting must have been significant for the formation and maintenance of social groups among the inhabitants of the coastal belt. Consequently, the porpoise may have had strong associations with coastal group identity and collective activities. 

A sense of seasonality must also have featured prominently in the perception of the porpoise among the communities involved in hunting it, given that the animal was present and accessible to hunters in the region only in summer. The warmer part of the year also offered the best conditions for pottery-making (vessels could be dried more quickly for firing), and, thus, the seasonal concurrence of the two tasks may have been a factor promoting their linkage through pottery decoration. It may be significant that mandibles—the bones utilized for this purpose—are absent among the faunal remains, except for a few small fragments. Presumably, they were being curated for this special use, rather than discarded along with other bones.

Because of these probable associations—with coastal environments, hunting collectives and the summer season—pottery decoration with the porpoise tooth stamp could have been heavily laden with meaning for the coastal porpoise-hunting groups, potentially conveying a self-identity founded upon the pattern of subsistence activities in these coastal areas. 

No other examples of pottery decoration with marine mammal bone or tooth impressions are known to the authors outside of the Baltic region. However, the widespread practice of impressing pottery with the serrated edges of *Cerastoderma* shells in the Neolithic period in south-western Europe (“Cardial pottery”) [51] and the use of the tips of marine snail shells for pottery decoration in Norway [52] (p. 111) might likewise be seen as reflecting the prominence of coastal resources in the world-view of the communities inhabiting these areas. 

However, how may we interpret the examples of pottery stamped with the teeth of this marine mammal that occur on inland sites? As discussed by Bērziņš and Dumpe [35], there are several possible explanations. 

The first possibility to consider is seasonal movement by groups from the inland lake basins to the coast for the purpose of exploiting seasonal resources (porpoise in the summer, but potentially also other resources at different times of the year), whereby pottery made at the coast during this season was being taken to inland sites along with other equipment and maybe used for transporting foodstuffs. However, the evidence from inland faunal assemblages and stable isotope data forces us to reject this hypothesis. The identified faunal remains from Neolithic sites in the Lubāns area do not include any marine species, which is unsurprising, considering the long distance from the seacoast. There is evidence of marine resources from sites in the Lake Burtnieks basin, located much closer to the sea. Thus, the faunal assemblage from Riņņukalns, in addition to mass remains of freshwater and migratory fish, also includes a minor proportion of marine fish [53], and seal teeth are represented among the tooth pendants from Neolithic graves at Zvejnieki [49] (pp. 144–147). However, the contact between this lake basin and coastal areas did not involve large-scale provision of marine food resources. This is demonstrated by the stable isotope data for human skeletal remains from the middle and second half of the 4th millennium BC at Riņņukalns and Zvejnieki, which indicate a diet based largely on freshwater fish, with no evidence for significant marine protein input in this period [53,54]. 

If we reject seasonal coast–inland mobility as an explanation for the occurrence of porpoise tooth decoration on pottery from inland sites, then we must address the possibility that the pottery vessels were among the items being transferred within the regional exchange network of this period. The exchange of tools used for making the porpoise tooth decoration, although it cannot be ruled out, seems markedly less plausible, since it would not account for the occurrence on inland sites of other characteristic traits of pottery from the coastal sites, as described above. The idea that pottery was being exchanged in the Neolithic does run contrary to the traditional view that early prehistoric pottery was locally made and, thus, serves to characterize the material culture of that particular group. On the other hand, the Neolithic archaeological assemblages from the eastern Baltic and neighboring regions testify to the existence of a highly developed system of exchange, in the frame of which amber ornaments, flint and chopping tools of Karelian metatuff were distributed across a very wide area [55,56]. Presumably, this exchange system also included a range of organic objects (e.g., skins of marine and land animals, honey and wax, plant materials) that have not been preserved. In this context, the idea that pottery could also have constituted an item of exchange seems entirely plausible, even if the focus may have been on other exchange items, the pottery vessels serving merely as containers for transportation. 

A third hypothesis to consider is that practices of exogamy contributed significantly to this pattern. If pottery-makers came to live with their spouses in inland lake basin communities, then they may have brought their own tools with them and may have applied their pottery production skills in the new place of residence. 

The latter two hypotheses are not mutually exclusive: both artefact exchange and exogamy could have functioned concomitantly in the maintenance of links between coastal and inland communities, and so we could potentially be dealing with the movement of potters (along with their decorating tools) as well as the movement of pottery vessels. 

Future petrographic and chemical analysis to characterize the clays and tempering materials of the region’s Neolithic pottery should permit the isolation of non-locally made ceramics. In addition, we require a better understanding of the social relationships governing the long-distance movement of various materials and finished artefacts in the Neolithic period in the eastern Baltic and neighboring areas. Moreover, this should enable a clearer contextualization of the pottery stamped with porpoise teeth.

## 5. Conclusions

The evidence from archaeological sites in the eastern Baltic region significantly enhances our understanding of the Holocene history of the harbor porpoise in the Baltic Sea basin. Seldom encountered in eastern Baltic waters in recent centuries, the porpoise was a common seasonal visitor, especially in the period ca. 3400–3000 cal. BC, when it had a major economic importance for the coastal communities of the Stone Age, as reflected by its representation in faunal assemblages. Additionally, the porpoise mandible was very commonly used as an instrument for decorating pottery, producing distinctive imprints that may have served as an expression of the value that coastal communities of that time attached to this animal. The occurrence of this kind of ornamentation also on pottery from inland sites is a very important marker of coast–inland contact, the nature of which remains to be further investigated.

## Figures and Tables

**Figure 2 animals-13-00909-f002:**
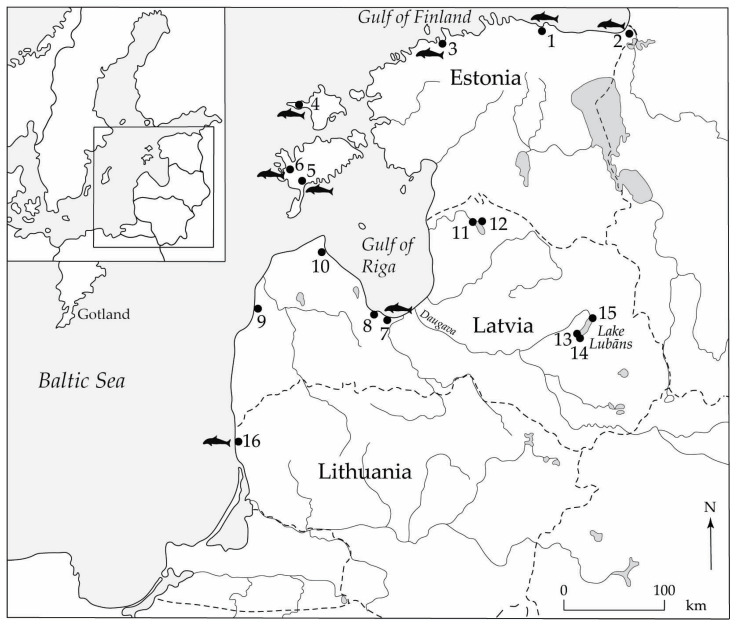
Neolithic sites with bone finds of porpoise (marked with porpoise symbol) and/or pottery with porpoise tooth impressions. 1—Kunda Lammasmägi; 2—Riigiküla; 3—Vabaduse Sq. in Tallinn; 4—Kõpu; 5—Naakamäe; 6—Loona; 7—Siliņupe; 8—Slocene; 9—Sārnate; 10—Pūrciems; 11—Riņņukalns; 12—Zvejnieki; 13—Malmutas grīva; 14—Suļka; 15—Piestiņa; 16—Šventoji.

**Figure 3 animals-13-00909-f003:**
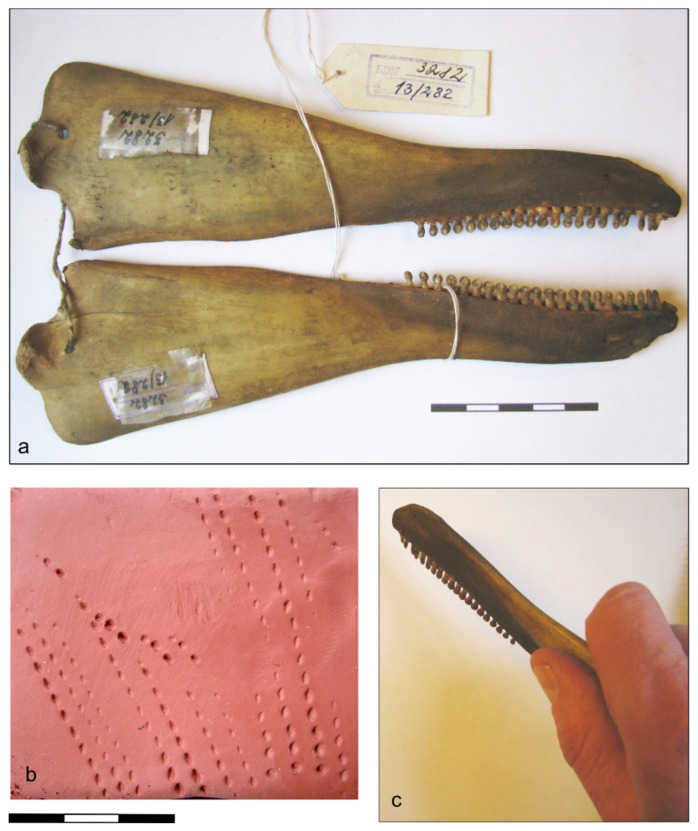
(**a**) Porpoise mandible in the collection of the Latvian Museum of Natural History used for the experiment; (**b**) tooth impressions obtained; (**c**) mode of holding the mandible for making impressions. Photos: V. Bērziņš.

**Figure 4 animals-13-00909-f004:**
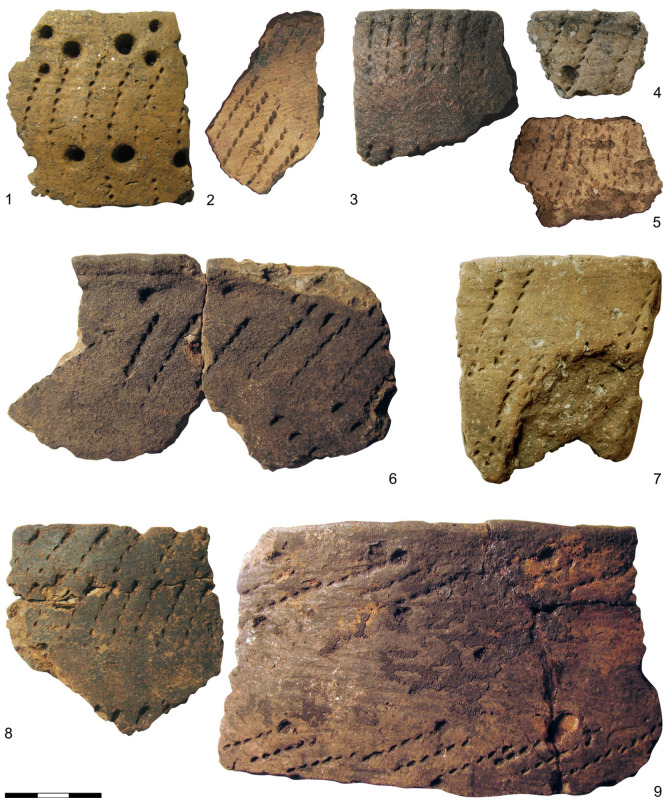
Pottery decorated with porpoise tooth impressions. 1—Naakamäe (AI 4211: 1531); 2—Piestiņa (VI 90); 3, 8—Siliņupe (VI 292); 4, 7—Riņņukalns (2018 excavation); 5—Malmutas grīva (VI 101: 19); 6, 9—Sārnate (A 11417: 206, A 11417: 313). 1– held in the Archaeological Research Collection of Tallinn University; 2, 3, 5, 6, 8, 9—held at the National History Museum of Latvia; 4, 7—temporarily stored at the Institute of Latvian History, University of Latvia.

**Table 1 animals-13-00909-t001:** Bone finds of harbor porpoise (*Phocoena phocoena*) from archaeological sites of Estonia, Latvia and Lithuania. NISP (number of identified specimens) is presented, and where available, the MNI (minimum number of individuals) is added (values separated by “/”).

Site	Collection No.	Skull	Mandibles	Atlas	Vertebrae ^1^	Limb Bones	Total
Kunda Lammasmägi							1
Riigiküla I	HI-51						3/2
Riigiküla III	AI 4198/AZ						1/1
Riigiküla IV	AI 8304/AZ						1/1
Loona	AI 4129/AZ				2		2/2
Naakamäe	AI 4211/AZ		4	16	431	19	470/35
Kõpu XI	AI 6106/AZ	5			1		6
Tallinn (Vabaduse Sq.)	AI 6917/AZ	33	1	6	77	15	132/8
Tallinn (6A Tatari St.)	AI 8636/AZ				1		1/1
Siliņupe		29			105	2	136
Šventoji 2B					1		1
Šventoji 3B					3		3

^1^ Includes a few ribs.

## Data Availability

Zooarchaeological data of Estonia will be available through the Estonian Archaeology dataset ArhIS (https://arhis.arheoloogia.ee; accessed on 15 January 2023). During the improvement of the dataset, the access is available only for administrators and registered users.

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
