# Peer review of "Natural History and Exploitation of the Harbor Porpoise (Phocoena phocoena Linnaeus, 1758) during the Neolithic (ca. 4000–2000 cal. BC) in the Eastern Baltic Region"

_animals, 2023, doi:10.3390/ani13050909_

Round 1
Reviewer 1 Report
Lembi Lõugas and Valdis Bērziņš paper entitled “Natural history and exploitation of the harbor porpoise (Phocoena phocoena) during the Neolithic in the eastern Baltic region” gives a review and new data on the topics under discussion. More specifically, a review of published archaeofaunal remains from this species in different Neolithic assemblages from Estonia, Latvia, and Lithuania are presented, some of which were subjected to revision by one of the authors. Small "experimentation" is also presented, where the mandible of a porpoise was used to reproduce imprints that were compared with a type of pottery decoration found mostly in coastal (but also scarcely inland) archaeological sites from the period being discussed. In fact, besides questions of exploitation and hunting strategies of porpoises, the authors include an interesting discussion on the use of this animal’s mandible tooth row beyond merely economic use, thus discussing the human-animal relations involving porpoises. The tables and figures accompanying the text are informative. The different sections are well-prepared and the bibliography is very complete and updated. The data, discussion, and conclusions are of interest, scientifically sound, and properly presented, thus following the standards of the journal. The manuscript is easily framed within the special issue topic. As such, I suggest the acceptance of the paper after minor revisions. A general appreciation of the different sections is presented below, followed by small comments and suggestions.
Simple Summary and Abstract: these are well-prepared and no suggestions are given to the authors on these sections.
Introduction: a general introduction to harbor porpoise distribution and characteristics, combining a large and different amount of sources is given (e.g., stable isotopes, DNA, ecology). This encompasses prehistoric to recent times, thus presenting an interesting historiographical perspective on part of the topics posteriorly discussed, with a clear description of the lack of information that occurs in specific areas or periods. The introduction clearly demonstrates the relevance of the analysis presented in this text.
Zooarchaeological material: the authors present the assemblages used in the revision, not only from a faunistic perspective but also discussing other types of evidence when available/needed to better understand the context (e.g., chronology).
Porpoise tooth impressions as decoration on pottery: the authors give a discussion on the use of porpoise teeth as a way of decorating pottery, with descriptions of pieces of evidence, their provenance, and chronology.
Discussion: this section is divided into two sub-sections. The existing evidence concerning porpoise exploitation is discussed with interesting chronological and environmental perspectives. The lack of a more robust chronological framework for some of the existing evidence is emphasized. The larger sub-section goes in depth in the discussion of the role of porpoises in human lifeways with some very interesting perspectives and hypotheses being presented. Although difficult to corroborate on some occasions considering available pieces of evidence, this is an interesting exercise where the caveats and limits of the data are also clearly recognized.
Conclusions: the main conclusions and hypothesis are succinctly and clearly presented.
Suggestions and modifications:
· Add a numeric chronology range in the title
· Line 94. Please change to “ca. 6000-3500 years ago”
· Figure 2. Add the north to the map
· Line 130. Please change to “Minimum number of individuals”
· Lines 134-137. Please check the use of several “;” that should be “,”
· Line 257. Change “c.” to “ca.”
· Line 298: Please remove the ()
· Line 303: please change “calibrated date of ca. 4080 BC” to “ca. 4080 cal. BC”
· Line 400: Please delete “by
· Currently, the use of mandibula/mandibulae is generally considered obsolete. I would advise changing this throughout the manuscript to the more commonly used form mandible/mandibles.
· I advise double-checking the use of different ways of presenting absolute chronologies in the paper to uniformize them when possible.
Author Response
Authors of the paper entitled “Natural history and exploitation of the harbor porpoise (Phocoena phocoena) during the Neolithic in the eastern Baltic region” are very thankful to reviewers for paying attention to the weak points in the manuscript, through which we can improve the article in a better way. We accepted the suggestions and modifications as follows:
- Add a numeric chronology range in the title – Numerical chronology added in the title.
- Line 94. Please change to “ca. 6000-3500 years ago” – The order of years changed.
- Figure 2. Add the north to the map – The north direction added and the map replaced.
- Line 130. Please change to “Minimum number of individuals” – changed
- Lines 134-137. Please check the use of several “;” that should be “,” – this was a question of formatting, and was not present in our submitted text. Now, it is fixed.
- Line 257. Change “c.” to “ca.” – changed
- Line 298: Please remove the () – removed
- Line 303: please change “calibrated date of ca. 4080 BC” to “ca. 4080 cal. BC” – changed
- Line 400: Please delete “by – deleted
- Currently, the use of mandibula/mandibulae is generally considered obsolete. I would advise changing this throughout the manuscript to the more commonly used form mandible/mandibles. – changed to mandible/mandibles throughout the manuscript
- I advise double-checking the use of different ways of presenting absolute chronologies in the paper to uniformize them when possible – accepted and unified where needed using “cal. BC”

Reviewer 2 Report
This was a very interesting paper that addresses a gap in literature identified by the authors. The discussion adds some interesting ideas about the cultural role of porpoises and the implications of inland finds of porpoise-impressed ceramics, which will hopefully initiate further discussion. I have only minor comments to make:
- In the title and first mention in the introduction please give the taxonomic authority for the species: Phocoena phocoena (Linnaeus, 1758). Similarly please give authorities for other taxa mentioned in the text.
-On line 52, should there be a capital P for Proper?
-On line 130 should read Minimum Number of Individuals
-Line 231 could simply say 'dates'
-Around line 313, it might be helpful to include something very brief about what caused the decrease in salinity
-Line 431 should read 'mandibula' I think
Figure 4 appears to have an additional number 5 on sherd 1.
Author Response
Review 2
Authors of the paper entitled “Natural history and exploitation of the harbor porpoise (Phocoena phocoena) during the Neolithic in the eastern Baltic region” are very thankful to reviewers for paying attention to the weak points in the manuscript, through which we can improve the article in a better way. We accepted the suggestions as follows:
- In the title and first mention in the introduction please give the taxonomic authority for the species: Phocoena phocoena (Linnaeus, 1758). Similarly please give authorities for other taxa mentioned in the text. – taxonomic authorities added to the Latin name when first mentioned in text
-On line 52, should there be a capital P for Proper? – Yes, we are consistently capitalizing it.
-On line 130 should read Minimum Number of Individuals – corrected
-Line 231 could simply say 'dates' – yes, corrected
-Around line 313, it might be helpful to include something very brief about what caused the decrease in salinity – sentence improved accordingly
-Line 431 should read 'mandibula' I think – all instances of „mandibula“ changed to „mandible“
Figure 4 appears to have an additional number 5 on sherd 1 – figure 4 replaced by new one without the extra number 5

Reviewer 3 Report
The article describes in a concise and appropriate way an analysis of the remains of sea mammals' species focusing on harbor porpoise from archeological sites in the Baltic region. The prevalence of these species today is low, but the archeological findings indicate that their prevalence was higher in the past. In addition, the results indicate that the exploitation of these species in ancient societies was not limited to food consumption but had also an artistic and decorative purpose, where pottery was decorated with mandibles marks. The importance of these decoration in ancient societies is also nicely demonstrated.
This work attempts to identify sea mammals remains and pottery with decorative marks of their teeth. Animal remains' details are well arranged in a summary table, with location details and counts. Yet, this summary data is missing for pottery remains, and should be included.
In summary, this is a good and publishable work, yet several elements of the work should be improved:
1) The introduction is well written, yet it would be helpful to add a figure of the major species tested.
2) the methods of material collection should be detailed – was it done based on manual collection or sediment sifting.
3) The results are well explained, yet it is better to include also a summary table with the pottery analysis details, and to indicate their purpose (eating, storage, or any other usage).
4) In the discussion, it may be helpful to describe examples of equivalent usage of sea mammals in non-Baltic societies.
Author Response
Authors of the paper entitled “Natural history and exploitation of the harbor porpoise (Phocoena phocoena) during the Neolithic in the eastern Baltic region” are very thankful to reviewers for paying attention to the weak points in the manuscript, through which we can improve the article in a better way. We accepted the suggestions as follows:
1) The introduction is well written, yet it would be helpful to add a figure of the major species tested. – Not sure whether we understood it right, but the word „figure“ was interpreted as „number“ and the number of cetacean species met in the eastern Baltic was added in the introduction section.
2) the methods of material collection should be detailed – was it done based on manual collection or sediment sifting. – The manual collection of archaeological bones was added in the text.
3) The results are well explained, yet it is better to include also a summary table with the pottery analysis details, and to indicate their purpose (eating, storage, or any other usage). - Phrase added, with reference, noting that vessels primarily used for cooking. A small supplementary table, S1, has been added, giving the statistics on the pottery with this kind of decoration and information about the excavations and collections, permitting some of the statistical data to be removed from the text.
4) In the discussion, it may be helpful to describe examples of equivalent usage of sea mammals in non-Baltic societies. - A short paragraph has been added, noting that no other examples of pottery decoration with marine mammal bone or tooth impressions are known to the authors outside of the Baltic region and also referring to impressions made with seashells in Neolithic pottery elsewhere in Europe, possibly indicating a similar prominence of coastal resources in the world-view.